# The oligomeric assembly of galectin-11 is critical for anti-parasitic activity in sheep (*Ovis aries*)

Dhanasekaran Sakthivel [1,2,3], Sarah Preston[2], Robin B. Gasser[4], Tatiana P. Soares da Costa[5], Julia N. Hernandez[6], Adam Shahine[1], M. D. Shakif-Azam[2], Peter Lock [7], Jamie Rossjohn[1,8], Matthew A. Perugini[5], Jorge Francisco González[6], Els Meeusen [2], David Piedrafita[1,2 ✉] & Travis Beddoe [1,3 ✉]

Galectins are a family of glycan-binding molecules with a characteristic affinity for ß-D-glycosides that mediate a variety of important cellular functions, including immune and inflammatory responses. *Galectin-11* (LGALS-11) has been recently identified as a mediator induced specifically in animals against gastrointestinal nematodes and can interfere with parasite growth and development. Here, we report that at least two natural genetic variants of LGALS-11 exist in sheep, and demonstrate fundamental differences in anti-parasitic activity, correlated with their ability to dimerise. This study improves our understanding of the role of galectins in the host immune and inflammatory responses against parasitic nematodes and provides a basis for genetic studies toward selective breeding of animals for resistance to parasites.

---

[1] Department of Biochemistry and Molecular Biology, Monash University, Clayton, VIC 3800, Australia. [2] School of Science, Psychology and, Sport, Federation University, Churchill, VIC 3842, Australia. [3] Department of Animal, Plant and Soil Science and Centre for Agri Bioscience (Agri Bio), La Trobe University, Bundoora, VIC 3086, Australia. [4] Faculty of Veterinary and Agricultural Sciences, The University of Melbourne, Bundoora, VIC 3010, Australia. [5] Department of Biochemistry and Genetics, La Trobe Institute for Molecular Science, La Trobe University, Bundoora, VIC 3086, Australia. [6] Instituto Universitario de Sanidad Animal, Faculty of Veterinary Medicine, Universidad de Las Palmas de Gran Canaria, Arucas, Spain. [7] Bioimaging Platform, La Trobe Institute for Molecular Science, La Trobe University, Bundoora, VIC 3086, Australia. [8] ARC Centre of Excellence in Advanced Molecular Imaging, Monash University, Clayton, VIC 3800, Australia. ✉email: david.piedrafita@federation.edu.au; t.beddoe@latrobe.edu.au

Galectins are proteins that specifically bind to β-galactoside sugars. Currently, 15 galectins are recognised in mammals and are encoded by LGALS genes. They include galectins-1, -2, -3, -4, -7, -8, -9, -10 and -12 of humans, and galectins-5 and -6 of rodents, and galectins-11, -14 and -15 of sheep and goats. Galectins have been found also in other species of mammals, as well as amphibians, birds, fish, nematodes, sponges and selected fungi[1]. They are soluble proteins that function within or outside of cells, and are present predominantly in the cytosol, nucleus, and extracellular matrix or in lymph. Although some galectins are secreted, they do not possess a classical (secretion) signal peptide, and it is not yet known how these molecules are secreted[1].

Mammalian galectins are known to mediate developmental processes, including cell differentiation, tissue organisation[2–4] and regulate immune homoeostasis[5,6]. They modulate the recognition and effector functions in innate immunity[7], and bind glycans on the surface of microorganisms and parasites[8–14]. Published evidence[9] shows that parasitic nematodes of the alimentary tract can upregulate galectin expression in selected tissues in stomach, small intestine and liver. In animals, for instance, nematode larvae induce increased expression of *galectin-11*, which is produced by epithelial cells and secreted into the gut lumen, coinciding with an eosinophil-biased inflammatory response characteristically associated with larval infection(s)[9,15,16]. As a functional analogue of galectin-1, galectin-11 is proposed to cross-link glycans in mucus to form a physical barrier against the parasite(s), as part of a pronounced inflammatory response[17,18]. The exacerbated expression of galectin-11 during challenge infection indicates that this molecule is intimately involved in innate and adaptive immune responses[15].

Despite this knowledge, the molecular biology and function(s) of galectin-11 are not understood. Gaining insights into the structural biology of this galectin is central to beginning to understand its function. In the present study, we explored the structure of galectin-11 of sheep (*Ovis aries*), and assessed whether and how amino acid alterations in this galectin and associated structural changes might affect its ability to interact with *Haemonchus contortus* (order Strongylida)—a major eukaryotic pathogen of the largest group of socioeconomically important nematodes of vertebrates[19]. This study found that, the presence of two genetic variants of LGALS-11 in sheep, and demonstrates that the oligomeric property of this galectin is critical for its anti-parasitic activity.

## Results

**Two isoforms of LGALS-11 detected in sheep**. To explore whether sequence variation in *galectin-11* (designated LGALS-11) occurs within *O. aries*, we sequenced this gene from 16 individuals (Merino breed) and deduced the amino acid sequences. Two distinct isoforms (1 and 2) represented 9 and 7 sheep, respectively (Fig. 1a). An alignment over 137 amino acids revealed that the two LGALS-11 isoforms differed at 15 amino acid positions (10.9%; Fig. 1a). We then explored the structure and function of these isoforms.

**Crystal structure of sheep LGALS-11**. First, we solved the crystal structure of LGALS-11 in apo form to 2.0 Å complexed with β-D-galactose at 2.4 Å (Table 1). Both the apo form and the ligand-bound crystals belongs to the space group of $P2_12_12_1$, comprising eight molecules in an asymmetric unit[20], suggesting that LGALS-11 can form higher order oligomers, like *galectin-1* (refs. [21,22]). The LGALS-11 structure has two anti-parallel, six-stranded (S1, 8–12; S2, 34–41; S3, 50–58; S4 59–67; S5, 70–72; S6a, 75–79; S6, 121–128) and five-stranded (F1, 16–27; F2, 88–96; F3, 97–104; F4, 106–113; F6, 129–137) β-sheets[23] with a typical *galectin*-fold

(Fig. 1b). The crystal structure of LGALS-11 has a folding pattern that is similar to prototype galectins-1, -2 and -10, suggesting a relatively conserved ligand-binding activity. LGALS-11 forms a homodimer via hydrogen bonds between residues L9 and S11 of monomers (i), and is inferred to have two unique cell attachment sites (called LDV and RGD) for integrin binding (ii). The asymmetric unit suggests that LGALS-11 can form higher order oligomers, like galectin-1 (refs. [21,22]).

**Glycan recognition by LGALS-11**. Complexing LGALS-11 with β-D-galactose allowed us to identify residues critical for glycan binding. This galactose bound directly to the residues present in the four adjacent strands (S3, S4, S5 and S6) on the concave face of one β-sheet (Fig. 1b). The glycan-binding groove expands from residues 50–74 (β4–β8 strands), and the carbohydrate-recognising residues (i.e., 54 R, 62 V, 64 N, 71 W and 74 E) within the carbohydrate recognition domain (CRD; Fig. 1c) are conserved, as expected of mammalian galectins[23].

In a structure-based alignment of these 16 LGALS-11 sequences, we identified a polymorphism at residue 9 (L/Q; Fig. 1a), within the glycan-binding domain and cell attachment domain. The glycan-binding domain (residues 53–74) of isoform 1 harboured highly hydrophobic residues (65–81; Fig. 2), consistent with other known galectins[1,24,25]. Variability in the glycan-binding groove (60 I/M, 65 T/A, 72 G/E and 75Q/E) of isoform 2 related to differences in hydrophobicity and electrostatic potential in the CRD (Fig. 2), suggesting that this isoform has a lower affinity for β-D-galactosides than isoform 1, and that it can interact with distinct glycans.

**Effect of sequence variation on quaternary structure**. To test the hypothesis that an amino acid change at position 9 alters the quaternary structure of LGALS-11, we first estimated the sedimentation velocities of recombinant isoforms 1 and 2. The $c(s)$ distributions (Fig. 3) at concentrations of 0.10, 0.30 and 0.90 mg/ml showed that isoform 1 is represented by a tetramer at all concentrations tested (sedimentation coefficient: 4.0 S), whereas isoform 2 is in a monomer–dimer equilibrium (sedimentation coefficients: 1.64 S and 2.5 S, respectively; Supplementary Table 1). To verify the tetrameric organisation of isoform 1, we then mutated residues L9 and S11 to alanine, produced a recombinant dimerisation mutant (DI-m) protein and assessed the sedimentation velocity of this protein. This DI-m protein exists as a monomer–dimer equilibrium (sedimentation coefficients: 1.1 S (monomer) and 1.7 S (dimer), respectively), similar to isoform 2 (Supplementary Table 1).

**Effect of quaternary structure on biological function**. To evaluate whether quaternary structure related to biological function, we exposed exsheathed third-stage larvae (xL3s) of *H. contortus* to the tetrameric isoform 1 or monomeric–dimeric isoform 2 (in solution) and assessed the effect of each isoform on the development of xL3s to the fourth-stage larvae (L4s) of *H. contortus*. In an initial experiment, galectin isoforms were tested at different doses between 0.009 and 2.5 mg/ml, and doses >0.75 mg/ml were shown to have biological effects after 24 h incubation. Hence further bioassays were tested at 1.0 mg/ml concentration of galectins. Most xL3s (85%) exposed to isoform 1 did not develop to L4s, whereas those exposed to isoform 2 developed normally (Fig. 4a). The incubation of (in vitro-raised) L4s with isoform 1 for 72 h reproducibly induced pronounced cuticular damage, a shrunken soma and an accumulation of internal granules in ≥95% of larvae, ultimately leading to larval destruction (Fig. 4b). By contrast, incubation of L4s with isoform 2 resulted in very limited internal or cuticular damage in <10% of larvae after 72 h (Fig. 4b).

**A.**

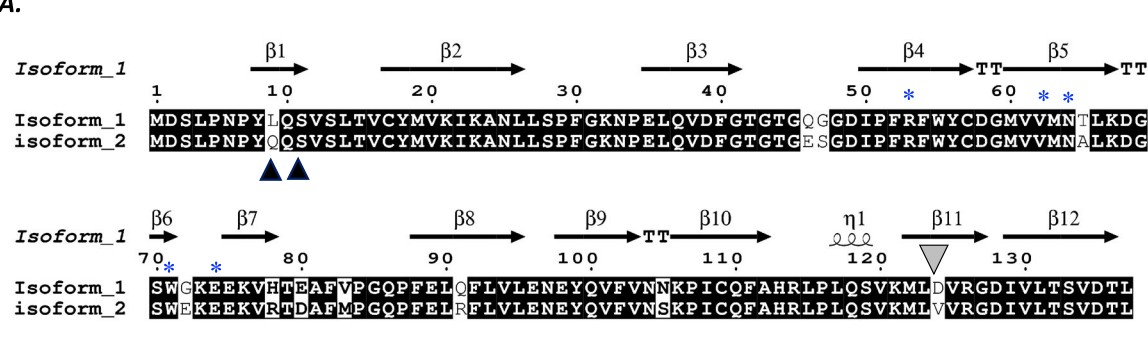

**B.**

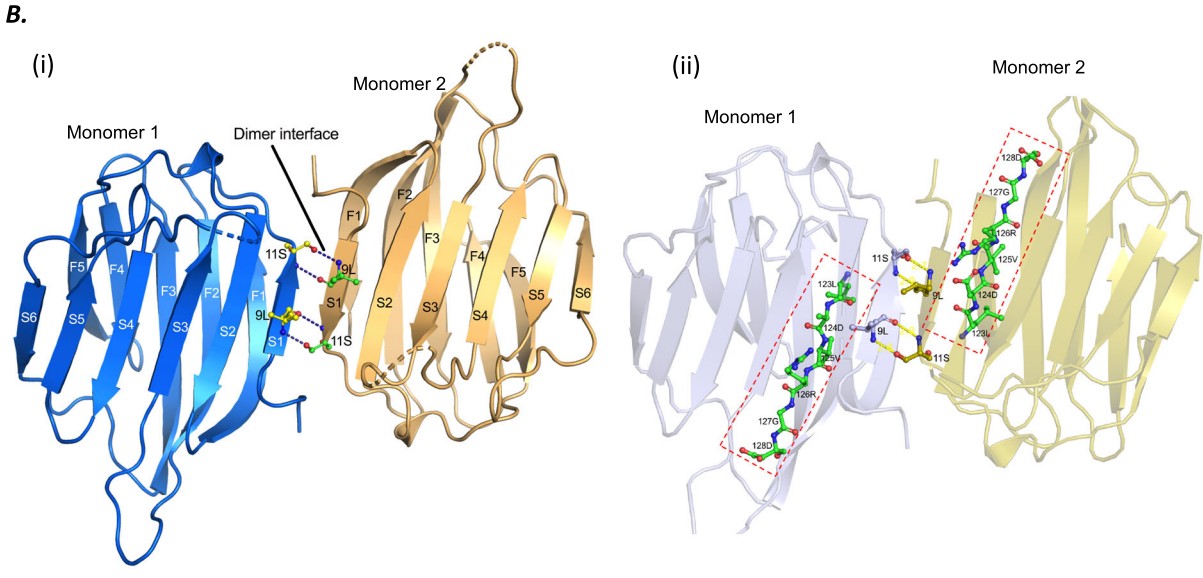

**C.**

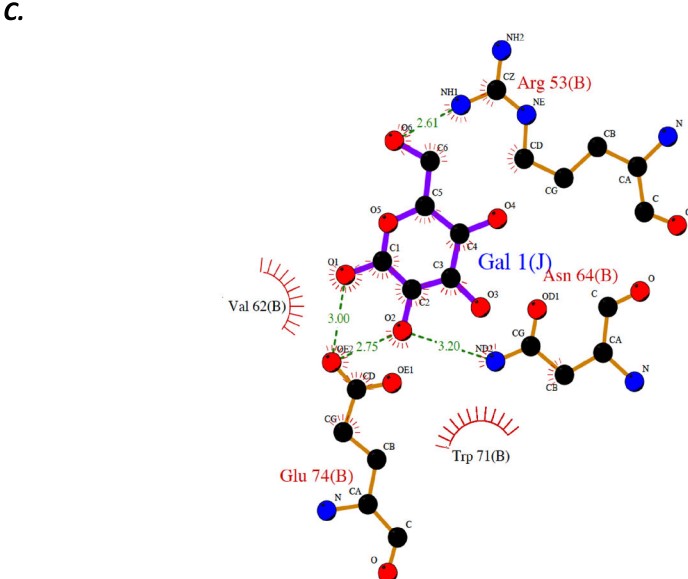

## Discussion

Here, we revealed the structure of LGALS-11 and showed that particular, single amino acid changes in this galectin can substantially alter this molecule's oligomeric assembly and its interaction with a parasitic worm (*H. contortus*) in vitro under well-controlled conditions. The findings from this investigation are likely to have important implications for understanding the fundamental molecular mechanisms of *galectin*-mediated parasite cytotoxicity.

The LGALS-11 crystal structure enabled the localisation of the precise residue variation in the dimer interface connecting β-strands of two monomers, the glycan-binding groove and the putative integrin-binding site. The residue substitution from a hydrophobic residue (leucine) in isoform 1 to a polar or charged

**Fig. 1 The crystal structure of LGALS-11. a** Structure-based sequence alignment of LGALS-11 natural isoforms. Conserved residues are indicated by white letters with black background, the dissimilar amino acids are indicated by black letters on a white background. The dimer interface is indicated by the black triangles (residue 9 L/Q). An amino acid difference in the putative integrin-binding site is denoted by the grey triangle (residue 124 D/V). Carbohydrate-recognising residues are indicated by blue asterisks. The secondary structures are indicated as beta (β) sheets with horizontal black arrows, turns with TT letters and helices with squiggles. **b** The crystal structure of LGALS-11, and residues involved in cell attachment. The dimer interface between monomers at residues L9 and S11 of the S1 β-strand via hydrogen bonds in a ball-and-stick representation; β-strands are labelled as S1–S6 and F1–F5 in both monomers (i). Predicted cell attachment residues (L123, D124, V125, R126, G127 and D128) of LGALS-11 in a ball-and-stick representation within the rectangle (red dashed line) (ii). **c** Residues in LGALS-11 are involved in glycan recognition. The residues R53, N64 and E74 of LGALS-11 make direct contact with β-D-galactose via hydrogen bonding (green dashes), and V62 and W71 make non-bonded contact. Image produced using PDBsum server (https://www.ebi.ac.uk/pdbsum/).

**Table 1 Data collection and refinement statistics of galectin-11 protein crystals.**

|  | Gal11 (apo) | Gal11 (complex) |
|---|---|---|
| Data collection |  |  |
| Space group | P2₁2₁21 | P2₁2₁2₁ |
| Cell dimensions |  |  |
| $a$, $b$, $c$ (Å) | 96.17, 127.45, 141.41 | 96.11, 127.43, 140.33 |
| α, β, γ (°) | 90, 90, 90 | 90, 90, 90 |
| Resolution (Å) | 38.38–2.00 | 47.17–2.40 |
| $R_{merge}$ | 0.20 (1.04) | 0.27 (1.27) |
| $I/\sigma I$ | 6.6 (1.9) | 7.3 (1.8) |
| Completeness (%) | 98.6 (95.4) | 99.4 (98.5) |
| Multiplicity | 7.9 (6.4) | 6.7 (6.4) |
| Refinement |  |  |
| Resolution (Å) | 33.38–2.0 (2.07–2.0) | 43.91–2.397 (2.48–2.39) |
| Total no. reflections | 915,448 (35,018) | 312,511 (28,539) |
| Total no. unique | 115,747 (11,080) | 68,269 (6720) |
| $R_{work}/R_{free}$ (%) | 19.2/23.8 | 16.04/20.91 |
| No. atoms |  |  |
| Protein | 8482 | 8519 |
| Ligand/ion | 30 | 60 |
| Water | 1478 | 1063 |
| B-factors | 16.26 | 22.26 |
| R.M.S. deviations |  |  |
| Bond lengths (Å) | 0.007 | 0.007 |
| Bond angles (°) | 0.96 | 0.89 |
| Ramachandran plot (%) |  |  |
| Favoured | 98.48 | 97.94 |
| Allowed region | 1.33 | 2.06 |
| Disallowed region | 0.19 | 0.00 |

Each data was collected from a single crystal.

residue (glutamine) in isoform 2 is likely to hinder dimer formation, and energetically favour contact with water rather than the complementary polar (serine) residue (S11) to form a homodimer. Through sedimentation velocity studies, we confirmed that the residue substitution in the dimer interface affects the homodimers and subsequent oligomerisation of LGALS-11. Dimerisation of prototype galectins (as is LGALS-11) are reported to be critical for their function; for example, the dimeric form of galectin-1 induces cell death (apoptosis), whereas the monomeric form does not[26,27]. As the quaternary structure and oxidation status of galectins alter ligand-binding and cross-linking properties[21,28,29], modulation of LGALS-11 dimerisation might obviate galectin–glycan lattice formation and glycan preference of isoform 2.

In contrast to most galectins that are constitutively expressed and have multiple biological functions, the expression of LGALS-11 in the gut epithelium appears only to be induced by parasites and pregnancy[15,16,30–33]. This information suggests a specific role of this molecule in these pathological and physiological processes. Studies of the protective immune response against *H. contortus*

have shown secretion of LGALS-11 in stomach (abomasal) mucus around the time that L3s moult to L4s, with LGALS-11 levels remaining elevated during infection with the adult stage of the pathogen[33]. Subsequent studies[9,17] showed that LGALS-11 could remarkably interfere with the moulting process to L4, and with the growth of this stage. The transition of L3s to L4s requires structural development facilitated by secretory processes[34,35] and takes place in the crypts of the abomasum, where the L4s are in close contact with the epithelial cells that are known to secrete LGALS-11[15]. Parasite stages, recovered in vivo from the abomasum of infected sheep, were found to be already covered with native galectin-11[9]. Viability loss and death of larvae in vivo were also associated with appearance of vacuoles and the cuticles shrunken away from their sheaths[34,36]. We have replicated these in vivo observations, with our in vitro studies and suggest that the mechanisms identified with recombinant galectin-11 in vitro can occur in vivo. In addition, we demonstrated that only the homotetrameric form of LGALS-11 inhibits *H. contortus* larval development, and causes direct damage to the surface of the L4, an indication of loss of viability and death[36]. No anti-parasitic activity was detected for isoform 2 or the dimerisation interface (DI-m) or carbohydrate recognition (CRD-m) mutant, indicating that extended oligomerisation or lattice formation via self-dimerisation and glycan recognition properties is critical for the anti-parasitic effect. Lattice formation by lectins is reported to be responsible for trapping and killing of free-living nematodes by nematophagous fungi[13,26,37], and a similar mechanism might be involved here.

Variable genetic resistance to *H. contortus* and related worms, both within and between breeds, is well known[38,39], but the molecular basis for this variation has not yet been established[38,40]. The discovery of at least two natural isoforms of LGALS-11 and their distinctly different anti-parasitic activities encourage future work to explore whether each indeed confers anti-parasite resistance or susceptibility. This could be done, for example, by conducting a large-scale study of allelic variation in LGALS-11 in populations of *H. contortus*-resistant and *H. contortus*-susceptible sheep[38], to test the hypothesis that quaternary structure of LGALS-11, determined by an alteration at position 9, confers natural resistance in sheep against *H. contortus*. Establishing an unequivocal link with resistance or susceptibility could aid the selective breeding of sheep for resistance to parasites, achieving natural parasite control, and circumventing substantial problems associated with anthelmintic resistance in *H. contortus* and related parasites[41]. To the best of our knowledge, for the first time this study demonstrates, the presence of two genetic variants of LGALS-11 and demonstrates that the oligomeric property of this galectin is critical for its anti-parasitic activity. Further studies are needed to explore whether such gene variation plays a critical role in the relative resistance of the host to parasite infection.

## Methods

**Molecular cloning of LGALS-11.** The cDNAs encoding LGALS-11 were PCR amplified from RNAs extracted from the stomach (abomasum) walls from 8–12-

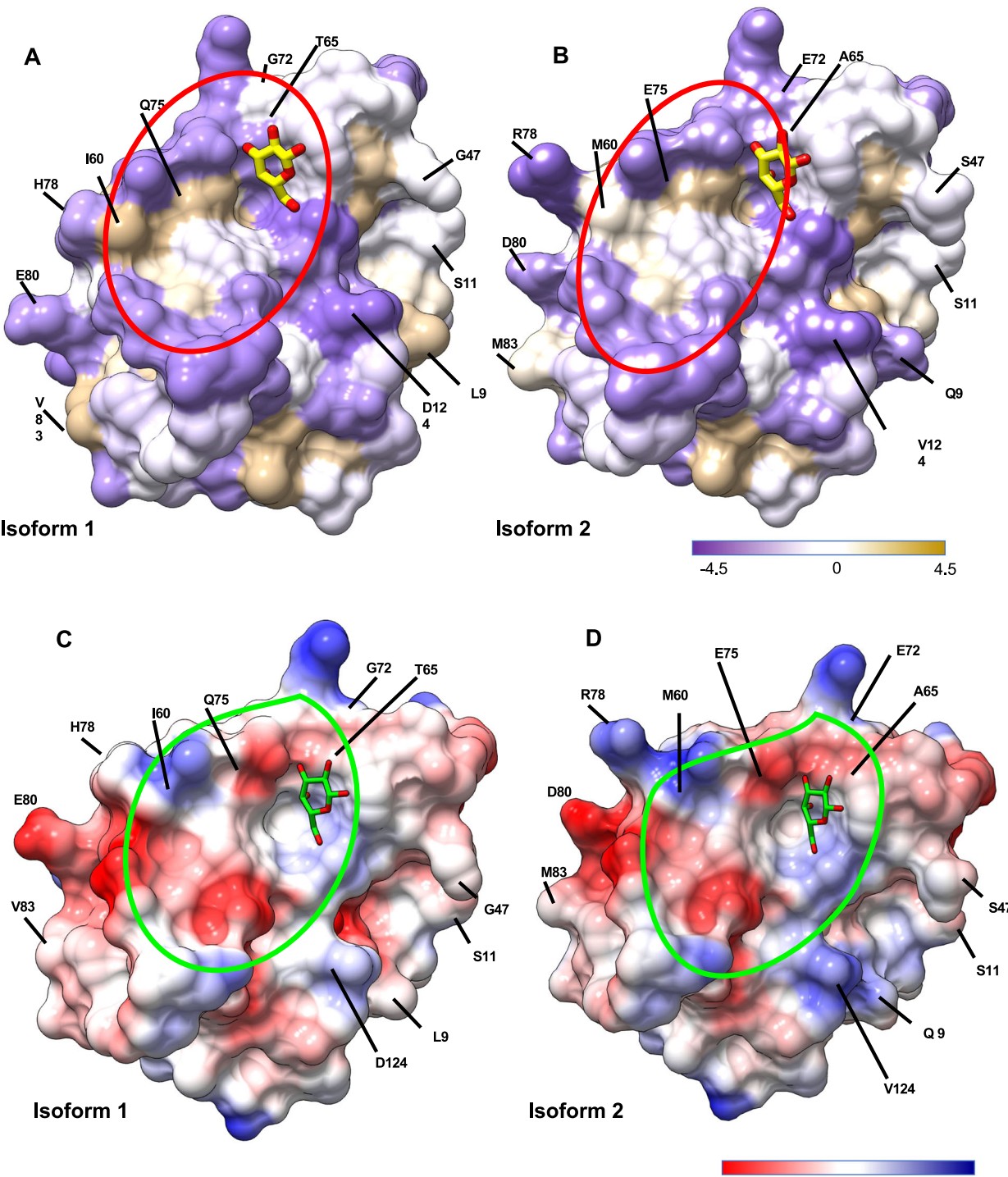

**Fig. 2 Visualisation of surface hydrophobicity, and electrostatic properties of the LGALS-11 complex with β-D-galactose.** Amino acid substitution in a surface representation of hydrophobicity (**a**, **b**) and electrostatic properties (**c**, **d**) of residues in isoforms 1 and 2 of LGALS-11 are labelled with a one letter code and position. The glycan-binding groove is highlighted and defined by dotted lines. The hydrophobicity properties of amino acids are indicated using the Kyte and Doolittle hydrophobicity scale; the most polar residues are in medium purple and the most hydrophobic residues are in tan in the surface representation. The electrostatic potential ranges from negative (red) to positive (blue). The surface properties of isoform 2 was derived by mutating the residues in the crystal structure of isoform 1 to isoform 2, using the COOT tool. The hydrophobicity and Coulombic electrostatic potential of both isoforms 1 and 2 were displayed using the surface colouring feature of the UCSF Chimera tool (v.1.10.2).

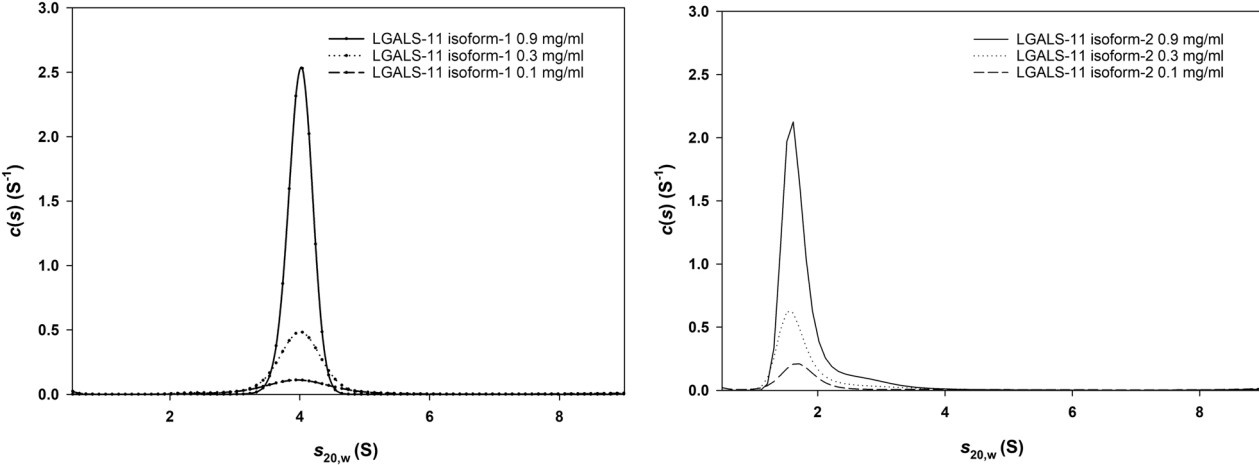

**Fig. 3 Sedimentation velocity analysis of LGALS-11 isoforms 1 and 2.** The continuous sedimentation coefficient [c(s)] distribution is plotted as a function of the standardised sedimentation coefficient at 0.10, 0.30 and 0.90 mg/ml, as indicated. Continuous size-distribution analysis was performed using the programme SEDFIT at a resolution of 200, with Smin = 0.5 S, Smax = 10 S, at a P-value of 0.95 for isoforms 1 and 2. Data are representative of three replicates.

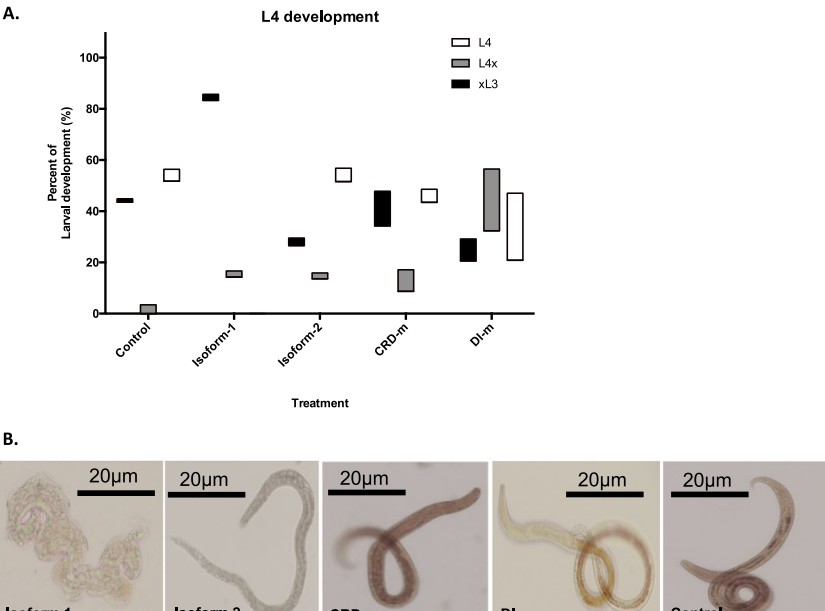

**Fig. 4 Anti-parasitic activity of LGALS-11 isoforms on *H. contortus*. a** The effect of LGALS-11 isoform 1 on *H. contortus* larval development. Exsheathed larval stages (xL3s) were incubated with LGALS-11 isoform 1 or 2, or carbohydrate recognition domain mutant (CRD-m) or dimer interface mutant (DI-m) for 7 days at 0.009–2.5 mg/ml concentration. L4 development was recorded microscopically. Control larvae were incubated for the same time only in the buffer used to resuspend individual LGALS-11 proteins. **b** Pathological changes observed in the L4 stage of *H. contortus* incubated with LGALS-11 isoform 1. Images showing representative phenotypes of L4 larvae following incubation with LGALS-11 isoform 1 or 2, or mutant CRD-m or DI-m for 3 days (1 mg/ml); control larvae were incubated for the same time only in the buffer used to resuspend individual LGALS-11 proteins. Extensive structural damage was observed in larvae incubated with isoform 1 at 1 mg/ml concentration. Scale bar = 20 μm.

month old Merino sheep (*O. aries*; n = 16) with mixed sex, and cloned into a modified pET-28 vector (m-pET-28) using an established method[20]. Clones representing all 16 individual animals were directly sequenced, and insert sequences verified by direct Sanger sequencing[42]. The similarity and secondary structures of sequences were inferred using the online tool ESPript-3.0 (ref. [43]). The LGALS-11 sequences (accession nos. MH069213 and MH069214) representing isoforms 1 and 2, respectively, were deposited in the NCBI database.

**Mutagenesis.** The cDNAs of two synthetic mutants of LGALS-11 were constructed (Bioneer Pacific, Australia). For mutant 1 (mutation in dimer interface designated as (DI-m)), an alanine was substituted for each of the leucine or glutamine (Q/L9A), and serine (S11A) residues in the dimer interface domain (Fig. 1a). For mutant 2 (mutation in CRD designated as CRD-m), an alanine was substituted for each of the residues P51A, R53A, W55A, V62A, N64A, W71AE74A

and K76A in the carbohydrate-recognising domain (Fig. 1a). Each of the two cDNAs was cloned into m-pET-28 using a ligation-independent approach[44], and insert sequences verified by direct sequencing[42].

**Recombinant expression and purification LGALS-11 isoforms.** Recombinant LGALS-11 isoforms 1 and 2, as well as the mutants DI-m and CRD-m were each expressed in *Escherichia coli* strain BL21 (DE3), purified using established methods[20] and stored at −80 °C until use.

**Measuring sedimentation velocity.** LGALS-11 isoforms 1 and 2 (0.10, 0.30 and 0.90 mg/ml) were each dissolved in Tris buffer (20 mM Tris-HCL pH 8.0, 100 mM NaCl, 0.5 mM TCEP pH 8.0) and their sedimentation velocities established using analytical ultracentrifuge (model XL-I, Beckman), equipped with a photoelectric optical absorbance system. Protein (380 μl) and reference (400 μl) samples were

loaded into a conventional double-sector quartz cell, mounted in a Beckman An-60 Ti rotor and centrifuged at $120,000 \times g$ at 20 °C. Data were collected at 280 nm in the continuous mode using a step size of 0.003 cm. Solvent density (1.00498 g/ml at 20 °C) and viscosity (1.002 cp at 20 °C), and estimates of the partial specific volume, $V\_$(0.744815 ml/g and 0.74269 ml/mg for isoforms 1 and 2, respectively, at 20 °C), were computed using the programme SEDNTERP[45]. Sedimentation velocity data at multiple time points were fitted to a continuous size-distribution model[46–48] using the programme SEDFIT[47]. The three independent experiments were performed.

**Crystallisation and X-ray diffraction analysis**. Recombinant LGALS-11 protein was purified, crystallised and diffraction data collected in Australian synchrotron as described[20]. The data were processed with MOSFLM and scaled using the programme AIMLESS of the CCP4 program suite. The initial phases for LGALS-11 were obtained by PHASER using the coordinates of human *galectin-10* (Charcot-Leyden crystal protein; PDB code: 1QKQ)[49]. Crystallographic refinement was done using PHENIX[50], and modelling was conducted using COOT v. 8.0 (ref. [51]). The PDB structure coordinates were validated using MOLPROBITY[52], and crystallographic images were drawn using PYMOL and UCSF chimera. The atomic coordinates and observed structural factors have been deposited (under codes 6N3R and 6N44, respectively) in the Research Collaboratory for Structural Bioinformatics PDB.

**Bioassay**. A well-established bioassay for *H. contortus* was used[9]. In this assay, each recombinant isoform ($n = 2$) and mutant ($n = 2$) of LGALS-11 was tested (in triplicate) on xL3 and L4 stages, using an irrelevant protein and Tris buffer (pH 7.4) as controls. This was done by adding individual recombinant proteins in Tris buffer (50 μl) at a serial concentration of 0.009 to 2.5 mg/ml (two steps) to wells each containing 300 xL3s or L4s in DMEM (pH 7.4), and incubated at 37 °C and 10% (v/v) $CO_2$. The motility of xL3s in individual wells was measured after 72 h, and the growth, development and morphology of L4s were assessed after 7 days.

**Ethics approval and consent to participate**. Handling of animals and experimental procedures were approved by the Monash University Animal Ethics Committee (Ethics # SOBSA/P/2009/44).

**Reporting summary**. Further information on research design is available in the Nature Research Reporting Summary linked to this article.

## Data availability

The information supporting the conclusions of this article is included in the article. The atomic coordinates and observed structural factors have been deposited (under codes 6N3R and 6N44, respectively) in the Research Collaboratory for Structural Bioinformatics PDB. The LGALS-11 sequences (accession nos. MH069213 and MH069214) representing isoforms 1 and 2, respectively, were deposited in the NCBI database. Source data underlying plots shown in Fig. 3 are provided in Supplementary Data 1. The authors declare that the data supporting the findings of this study available within the paper, any other relevant data in this study available from the corresponding authors on reasonable request.

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

## Acknowledgements

We thank staff at the Monash Macromolecular Crystallisation Facility (MMCF) for assistance with crystallisation, and at the Australian Synchrotron for support with X-ray data collection. We are grateful to Dr. Dene Littler for initial support with crystal structure analysis. DS acknowledges support from the Department of State Development, Business and Innovation (DSDBI), Victoria, Australia and Australia-India Institute (AII). DS acknowledges the financial support from the Australian Society for Parasitology to attend CCP4 crystallography workshop. R.B.G.'s research was funded through grants from the Australian Research Council (ARC), the National Health and Medical Research Council (NHMRC) of Australia, Yourgene Bioscience Taiwan, and Melbourne Water Corporation.

## Author contributions

D.S., D.P. and T.B. conceived this study; D.S., S.P., T.P.S.d.C., J.N.H., M.D.S.-A. and P.L. performed the research; D.S., S.P., A.S., J.R., M.A.P., J.F.G., E.M., D.P., R.B.G. and T.B. analysed data; D.S., D.P., R.B.G., E.M. and T.B. wrote the paper.

## Competing interests

The authors declare no competing interests.
