## [Peer Review File · Communications Biology]

Reviewers' comments:

Reviewer #1 (Remarks to the Author):

The manuscript describes a detailed structural and functional analysis of galectin-11, a galectin which has previously been shown to be highly upregulated in both sheep and cattle following nematode infections. The manuscript is well written and the results clearly presented and discussed. My comments are basically limited to bioassay performed to investigate the direct effect of the galectin on larval development. First, neither the text or the figure provides any information on whether there was a dose dependent effect. Although the materials and methods section described the use of the galectins at several concentrations, it is unclear at which concentration the effects were seen. I believe this is crucial information. Second, related to my first point, whether the galectin has the same activity in vivo as observed in the in vitro assay could also be dependent on the in vivo concentration of the galectin in the abomasal mucosa. This aspect is unfortunately not discussed in the manuscript. The discussion is written in such a way that the in vivo activity of galectin-11 seems to be clear-cut. I am personally not sure one can actually make this conclusion. Galectin-11 has shown to be relatively quickly upregulated following infection, long before true signs of protection are being observed. I therefore feel that the discussion should be a bit more nuanced.

Reviewer #2 (Remarks to the Author):

The manuscript "The oligomeric assembly of galectin-11 is critical for anti-parasitic activity" points out the association between the structure of galectin-11 and the functional activity tested on the development of the different forms of the *Haemonchus contortus*, which is one of the most economically important parasites of small ruminants worldwide. For thus, the authors found by sequencing the LGALS-11 gene from 16 individuals of sheeps, two different isoforms of this lectin. Although the work is a new contribution of the authors to the study of galectin-11 and its activity in nematode infection, some major concerns have to be addressed.

1) The two isoforms, 1 and 2, have been found in 9 and 7 different sheeps, so one should be inferred that both do not co-exist in the same individual. That is correct? And could be the chance that different isoforms could be expressed in different tissue or organs of the same individual? Which tissue was used to isolate the isoforms in the work? And so, the author did not show as supplementary the production and purification of both isoforms and the mutants. This reviewer considers that these data would improve the manuscript.

2) To my knowledge, the glycan specificity of galectin is still unknown, so it is possible that galactose-D could not be the best molecule to analyze its interaction the aminoacidic residues? In there, authors show the crystal structure of isoform 1, having both as recombinant. Any reason for that choice?

3) Crystal structure was carried out with the isoform 1 bound to D-galactose. What tag was used to produce it? (Please clarify in the text) Affect it to the structure and interaction with the glycan? Any reason to explain why authors did not perform the crystal structure of the other isoform?

4) Mutation on non residue L9 and S11 only modified the quaternary structure? Could it affect glycan specificity and this, alter the biological function of the lectin?

5) In a previous work (*International Journal for Parasitology*, 2015), authors demonstrated that galectin-11 alters the development of the t of the infective xL3s into L4 of *H. contortus* but also the viability of this form, so it would improve the manuscript if the author could show the role of each isoform on parasite growth.

Minor comments:

1) Add references on line 78.

2) Erase also on line 80 just because modulation of innate immune response is part of regulation of immune homeostasis mentioned before.

- 3) Isoform 2 has lower affinity for which glycan? Please clarify (line 133)
- 4) It would be better if results from the quaternary structure would be in the ítem Structure instead of with Biological function. Same for the figure (4) showing these data.
- 5) Please clarify in the text and also in figure legend the meaning of DI-m and CRD-m, because as manuscript structure, both abbreviations are at M&M at the end of the text.

Reviewer #1 (Remarks to the Author):

Q1. First, neither the text or the figure provides any information on whether there was a dose dependent effect. Although the materials and methods section described the use of the galectins at several concentrations, it is unclear at which concentration the effects were seen. I believe this is crucial information.

We have amended the manuscript text (line 158 – 160) and figure legends (lines 490-512) to reflect the dose used.

Q2. Second, related to my first point, whether the galectin has the same activity in vivo as observed in the in vitro assay could also be dependent on the in vivo concentration of the galectin in the abomasal mucosa. This aspect is unfortunately not discussed in the manuscript.

Q3. The discussion is written in such a way that the in vivo activity of galectin-11 seems to be clear-cut. I am personally not sure one can actually make this conclusion. Galectin-11 has shown to be relatively quickly upregulated following infection, long before true signs of protection are being observed. I therefore feel that the discussion should be a bit more nuanced.

We have amended the manuscript text (line 189-199) in the discussion to address these points to suggest our in vitro observations (Q2) are likely (Q3: nuanced suggestion) to occur in vivo: “The transition of L3s to L4s requires structural development facilitated by secretory processes (Veglia, 1915; Sommerville, 1966) and takes place in the crypts of the abomasum where the L4s are in close contact with the epithelial cells which are known to secrete galectin-11 (Dunphy et al., 2000). Parasite stages, recovered in vivo from the abomasum of infected sheep, were found to be already covered with native galectin-11 (Preston et al., 2015). Viability loss and death of larvae in vivo were also associated with appearance of vacuoles and the cuticles shrunken away from their sheaths (Veglia, 1915; Rios-de alvarez et al., 2012). We have replicated these in vivo observations, with our in vitro studies and suggest that the mechanisms identified with recombinant galectin-11 in vitro can occur in vivo.”

Reviewer #2 (Remarks to the Author):

Q1. The two isoforms, 1 and 2, have been found in 9 and 7 different sheep, so one should be inferred that both do not co-exist in the same individual. That is correct?

Q2. And could be the chance that different isoforms could be expressed in different tissue or organs of the same individual?

The authors have sequenced from multiple sheep using multiple clones from the same animal and did not find any co-existence of isoforms 1 and 2 in sheep. The authors have initially verified this hypothesis by testing sheep skin tissue and abomasum tissues and found the same isoforms in both the organs. However, the authors have not categorically stated in the manuscript that isoforms could not co-exist in different tissues as this would require extensive future work, well beyond the scope of this manuscript.

Q3. Which tissue was used to isolate the isoforms in the work?

The cDNAs encoding LGALS-11 were PCR-amplified from RNAs extracted from the stomach (abomasum) walls (Line 227 - 228).

Q4. And so, the author did not show as supplementary the production and purification of both isoforms and the mutants. This reviewer considers that these data would improve the manuscript.

The entire expression, purification and crystallization of galectin-11 isoform-1 is referenced in the manuscript (Line 111, 229, 247, 262; Reference #20) as our group has previously published this work.

Reference: Sakthivel, D., Littler, D., Shahine, A., Troy, S., Johnson, M., Rossjohn, J., Piedrafita, D., and Beddoe, T., (2015), Cloning, expression, purification and crystallographic studies of galectin-11 from domestic sheep (Ovis aries), Acta Cryst., F69, 993-997

Q5. To my knowledge, the glycan specificity of galectin is still unknown, so it is possible that galactose-D could not be the best molecule to analyze its interaction the aminoacidic residues?

The authors agrees with the reviewer, that glycan specificity of galectin-11 is unknown. However, galectin dogma is based on the definition as a family of β -galactoside-binding proteins characterized by a unique CRD sequence motif (present in galectin-11). Therefore, by definition, galactose is the best sugar to define the sugar-binding site. Parasite stages, recovered in vivo from the abomasum of infected sheep, found to be already covered with native galectin-11, which could be removed in a carbohydrate-specific manner with galactose (Preston et al. 2015; ref 9 in manuscript).

Q6. In there, authors show the crystal structure of isoform 1, having both as recombinant. Any reason for that choice?

Q8. Any reason to explain why authors did not perform the crystal structure of the other isoform?

Initial crystallization trials were carried out using isoform 1 as at that stage we were not aware of the polymorphism in galectin-11. It took over 18 months to achieve crystals that would diffract to higher enough resolution for structure determination. After the structure of isoform-1 was determined, we investigated whether there were other isoforms of galectin-11.

Crystallization of isoform-2 is on-going and will form the basis for a further publication.

Q7. Crystal structure was carried out with the isoform 1 bound to β -D-galactose. What tag was used to produce it? (Please clarify in the text) Affect it to the structure and interaction with the glycan?

Q9. Mutation o non residue L9 and S11 only modified the quaternary structure? Could it affect glycan specificity and this, alter the biological function of the lectin?

The recombinant galectin variants were produced having a HRV 3C-cleavable N terminal hexahistidine tag. The authors found that the removal of the his-tag, did not affect its glycan binding. The entire expression, purification and crystallization of galectin-11 isoform-1 has been published and referenced in a previous manuscript as detailed in Q4.

We tested the isoform on immobilized-galactose column as reported in Sakhivel, D., et al. (2015), Acta Cryst., F69, 993-997 and it did not affect glycan binding (ref 20 of manuscript).

Q10. In a previous work (International Journal for Parasitology, 2015), authors demonstrated that galectin-11 alters the development of the of the infective xL3s into L4 of *H. contortus* but also the viability of this form, so it would improve the manuscript if the author could show the role of each isoform on parasite growth.

This extensive work, along with the crystallization studies (Q8) is on-going and will form the basis for a further publication.

Minor comments:

Q11. Add references on line 78.

Reference included as requested.

Q12. Erase also on line 80 just because modulation of innate immune response is part of regulation of immune homeostasis mentioned before.

Word “also” removed in line 80.

Q13. Isoform 2 has lower affinity for which glycan? Please clarify (line 133)

“β-D-galactosides” in line 133 added, as requested for clarification.

Q14. It would be better if results from the quaternary structure would be in the item Structure instead of with Biological function. Same for the figure (4) showing these data.

The authors have amended the figure numbering as requested for the quaternary structure and biological functions.

Q15. Please clarify in the text and also in figure legend the meaning of DI-m and CRD-m, because as manuscript structure, both abbreviations are at M&M at the end of the text.

The authors have provided the details for abbreviated form of DI-m and CRD-m in line 237, 239 and 497-503, as requested.

REVIEWERS' COMMENTS:

Reviewer #2 (Remarks to the Author):

I accepted the manuscript after the authors' revision and edition.